# A lysine ring in HIV capsid pores coordinates IP6 to drive mature capsid assembly

**Nadine Renner**[1☯], **Donna L. Mallery**[1☯], **K. M. Rifat Faysal**[2], **Wang Peng**[2], **David A. Jacques**[2], **Till Böcking**[2], **Leo C. James**[1]*

**1** MRC Laboratory of Molecular Biology, Cambridge, United Kingdom, **2** EMBL Australia Node in Single Molecule Science and ARC Centre of Excellence in Advanced Molecular Imaging, School of Medical Sciences, UNSW Sydney, Australia

☯ These authors contributed equally to this work.

* lcj@mrc-lmb.cam.ac.uk

**Data Availability Statement:** The authors confirm that all data underlying the findings are fully available without restriction. All data is provided and made available in the manuscript and

## Abstract

The HIV capsid self-assembles a protective conical shell that simultaneously prevents host sensing whilst permitting the import of nucleotides to drive DNA synthesis. This is accomplished through the construction of dynamic, highly charged pores at the centre of each capsid multimer. The clustering of charges required for dNTP import is strongly destabilising and it is proposed that HIV uses the metabolite IP6 to coordinate the pore during assembly. Here we have investigated the role of inositol phosphates in coordinating a ring of positively charged lysine residues (K25) that forms at the base of the capsid pore. We show that whilst IP5, which can functionally replace IP6, engages an arginine ring (R18) at the top of the pore, the lysine ring simultaneously binds a second IP5 molecule. Dose dependent removal of K25 from the pore severely inhibits HIV infection and concomitantly prevents DNA synthesis. Cryo-tomography reveals that K25A virions have a severe assembly defect that inhibits the formation of mature capsid cones. Monitoring both the kinetics and morphology of capsids assembled in vitro reveals that while mutation K25A can still form tubes, the ability of IP6 to drive assembly of capsid cones has been lost. Finally, in single molecule TIRF microscopy experiments, capsid lattices in permeabilised K25 mutant virions are rapidly lost and cannot be stabilised by IP6. These results suggest that the coordination of IP6 by a second charged ring in mature hexamers drives the assembly of conical capsids capable of reverse transcription and infection.

## Author summary

HIV protects its RNA genome while copying it into DNA by carrying out reverse transcription inside its capsid. This is accomplished by importing nucleotides through highly charged pores at the centre of each capsid multimer. These pores contain two rings of positively charged residues–R18 and K25 –but assembling capsids with these features is challenging because they are intrinsically destabilising. Here we show that the metabolite IP6 coordinates both residues within the pore to drive the assembly of stable capsids capable of nucleotide import. R18 or K25 mutants lose infectivity and the ability to synthesise

supplementary data. In addition, the structural model and data are deposited in the PDB database with code 6R6Q.

**Funding:** LCJ, NR and DLM was supported by the MRC (UK; U105181010), a Wellcome Trust Investigator Award (200594/Z/16/Z) and a Wellcome Trust Collaborator Award (214344/A/18/Z). TB, KMRF, DAJ and WP were supported by the NHMRC (Australia; APP1100771 and APP1158338). The use of facilities in the Structural Biology Facility within the Mark Wainwright Analytical Centre –UNSW is funded in part by the Australian Research Council Linkage Infrastructure, Equipment and Facilities Grant: ARC LIEF 190100165.The funders had no role in study design, data collection and analysis, decision to publish, or in the preparation of the manuscript.

**Competing interests:** The authors have declared that no competing interests exist.

DNA but have differing assembly phenotypes. Mutant K25A is unable to undergo efficient capsid assembly, while replacing K25 with a neutral polar residue partially restores assembly but not infectivity. We propose that IP6-driven assembly is conserved by HIV not because it is the only way to build a capsid, but because it allows the construction of a capsid with a charged pore that can import nucleotides.

## Introduction

HIV builds its capsid in a two-step process. As the virus assembles at the plasma membrane, the Gag polyprotein first forms a lattice comprised of immature hexamers. Upon viral budding, the HIV protease cleaves Gag in multiple places in a process called maturation. The processed CA protein then assembles into its final structure, a conical capsid made up of hexamers and 12 pentamers. While the two hexamers, immature and mature, are structurally distinct they share a highly unusual feature; rings of positively charged residues lining a central cavity. In the case of the immature hexamer there are two lysine rings, K158 and K227[1], while in mature hexamers this is an arginine ring R18 and a lysine ring K25[2]. These would not be expected at the centre of a protein assembly due to the destabilising influence of clusters of positive charge, an effect that has been shown to be the case for R18[2].

Recent work suggests these features are encoded for a specific purpose, namely the ability to bind negatively charged small molecules such as inositol hexakisphosphate (IP6). IP6 is thought to drive the assembly of immature hexamers by coordinating the two lysine rings. It greatly increases the efficiency of immature viral-like particle (VLP) formation in vitro[1], whilst analysis of HIV virions reveals that IP6 is incorporated into particles at levels that are reduced when K227 is mutated[3]. IP6 also promotes the in vitro assembly of mature VLPs and stabilises mature hexamers via interaction with R18[1, 4]. Importantly, depletion of cellular IP6 upon knockout of the kinases (IPPK or IPMK) that are responsible for its synthesis, results in a reduction in HIV production. Conversely, IP6 depletion in target cells does not impact infection[3]. Nevertheless, IP6 is critical for HIV infectivity as lysine mutants that reduce packaging are poorly infectious, as are mutants of R18[2, 3].

Despite the above similarities, mature hexamers have properties that are lacking in their immature counterparts. On the outside face of mature hexamers, there is a β-hairpin structure which isomerises between different conformations[2] that either open or close access to the chamber where IP6 binds[4]. This feature means that the chamber is actually a pore, potentially allowing molecules to pass through the centre of mature hexamers and into the interior of an assembled HIV capsid. Importantly, IP6 is not the only molecule that binds to the R18 ring inside the pore and dNTPs are also recruited with a remarkably fast on-rate. We have proposed that this may allow HIV to recruit nucleotides into the interior of the capsid to fuel DNA synthesis[2]. This is essential because the virus has to reverse transcribe its RNA genome into DNA before it can integrate into the host genome. Reverse transcription (RT) and, thus, productive infection are closely correlated to capsid stability and its loss, a process called uncoating. [5–7]. Recently it has been shown that the HIV capsid remains largely intact until it enters the nucleus. Reverse transcription is then completed in the nucleus prior to uncoating [8, 9]. Thus, there must be a mechanism to allow dNTPs to enter the capsid to allow reverse transcription.

It is currently unclear whether none, one or both IP6 and nucleotide binding to mature hexamers is important in HIV biology. There are several reasons why unpicking this is not straightforward. It is difficult to probe a requirement for IP6 in viral maturation without also

impacting on immature assembly because altering incorporation can only be done by manipulating binding to the immature hexamer. Conversely, mutating R18 in the mature capsid destroys both IP6 and dNTP binding, making it difficult to dissect these as separate activities [2, 4].

In this study, we sought to understand more about the role of IP6 and dNTP binding in the mature capsid by focusing on the lysine ring at K25. Previous work has suggested that mutation of this residue impacts the production of infectious HIV virions. Mutant K25I virions were non-viable and production was severely reduced, although budding virions appeared morphologically similar to wild-type [10], while in another study K25A/R/E/C/L Gag mutants were seen to assemble with comparable efficiency [11]. Meanwhile, electric potential calculations along the hexamer pore axis have suggested that K25 may promote nucleotide translocation into the capsid by creating a lower energy potential barrier for import than dissociation back out of the pore [12]. Mutant K25A was found to increase the energy barrier and encourage dNTP movement out of the capsid. However, a second molecular dynamics study found the opposite, with mutation K25A reducing rather than increasing the energy barrier to nucleotide import [13]. Thus, while K25 is implicated in capsid assembly, virion production, infection and nucleotide import there is little consensus on its role or mechanism of action. Here we show that K25 is capable of recruiting a second inositol phosphate molecule, in addition to that bound by R18. Sequentially replacing each K25 within the hexamer with alanine leads to a dose dependent decrease in infection and RT. Purified K25A virions are also less able to utilize dNTPs to synthesize DNA in vitro. These data are supportive of a role for K25 in importing dNTPs to allow RT and promote infection. However, analysis of HIV capsids within virions by Electron cryotomography (CryoET) and single molecule total internal reflection fluorescence (TIRF) microscopy reveals that K25 mutants have a severe assembly and stability defect. In vitro assembly experiments confirm that K25 is required for efficient IP6-driven assembly of mature capsid cones. On the basis of these results, we propose that K25 uses IP6 to promote mature capsid assembly. Nevertheless, the inability of those K25A capsids that do assemble to undergo efficient reverse transcription suggests it is also required for dNTP import.

## Results

During attempts to crystallise the HIV hexamer in complex with IP6, we noted the presence of additional electron density for a possible second molecule in several datasets. However, the data were too weak to allow an additional ligand to be built with confidence. This is partly an intrinsic problem of the axial phosphate in IP6, which as a result of the 6-fold symmetry within the pore gives averaged data that makes the orientation of the ligand ambiguous. To circumvent this problem and obtain a definitive answer of whether a second IP molecule can bind at the pore we screened crystals complexed with IP5, which only contains equatorial phosphates. We were able to solve a hexamer structure in which there was clear density for two molecules of IP5; one molecule co-ordinated by R18 and a second by K25 (Fig 1A and 1B and Table 1). The two IP5s are located at the centre of the pore (Fig 1C) and adopt an approximately planar conformation, with one stacked above R18 and one above K25 and separated by ~10–12 Å. The symmetrically equivalent copies overlay closely, explaining the clear density. When viewed in a surface representation, it is clear that one IP5 molecule is trapped within the chamber bound by the β-hairpin at one end and R18 at the other, while the second molecule is located between R18 and K25 (Fig 1D). Based on this structure and a previous structure of hexamer with one bound IP6 (6ES8), we modelled a second IP6 into the pore (S1A Fig). The axial phosphates of both molecules are readily accommodated without steric hindrance. While our

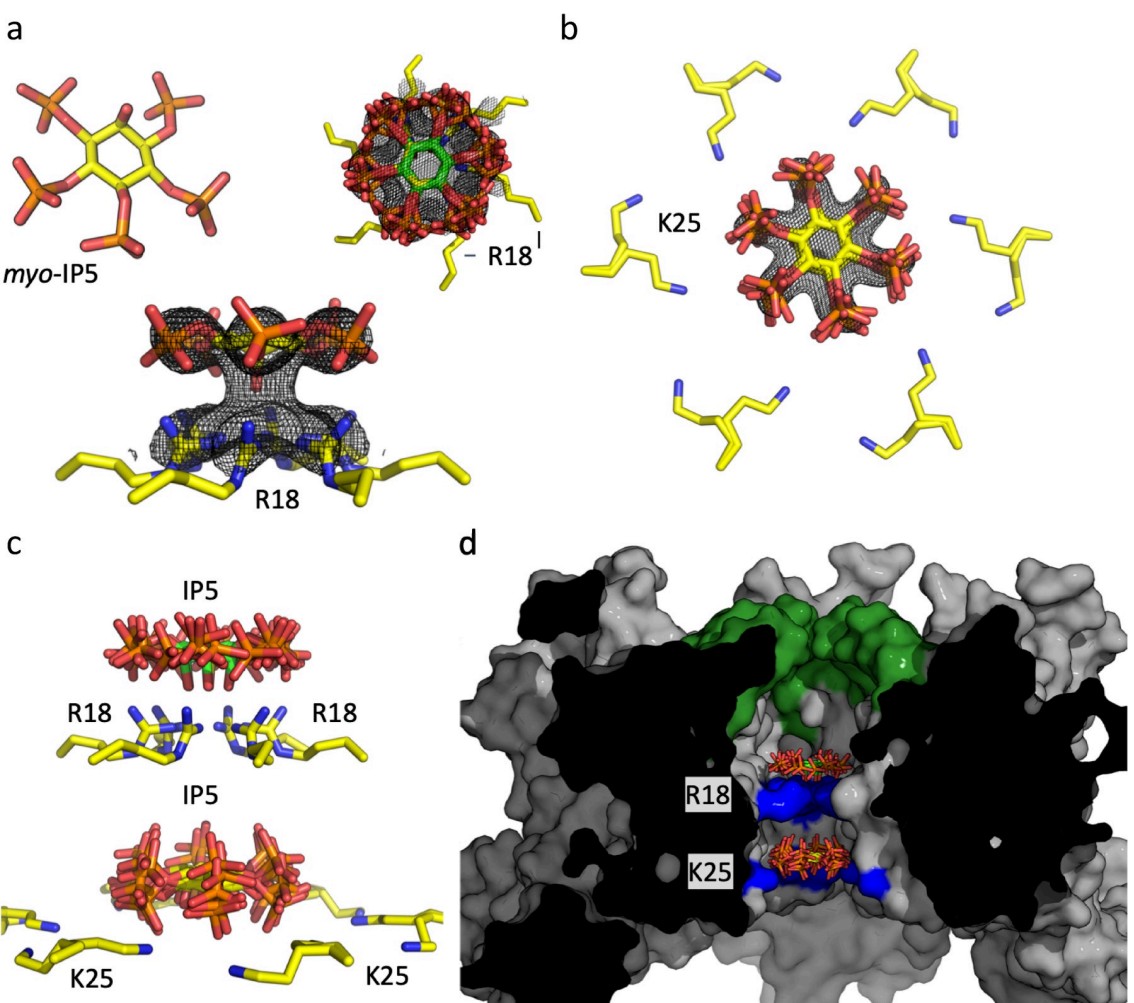

**Fig 1. Structure of IP5 bound to HIV CA hexamer.** (a) (left) Structure of myo-IP5 ligand. $F_o − F_c$ omit density (mesh) contoured at 2.0σ centered on IP5 bound to R18 and viewed down the 6-fold axis (right) and from the side (below). (b) $F_o − F_c$ omit density (mesh) contoured at 2.0σ centered on a second IP5 molecule next to K25 (two rotamer side chains are shown). All six-symmetry equivalent IP5 molecules are shown. (c) View showing the two IP5 molecules, one binding above R18 and one above K25. Note the second IP5 molecule is located closer to the K25 ring than R18. (d) Cross section through the hexamer, showing the central chamber where the IP5 molecules are bound. The β-hairpin is shown in green and the location of R18 and K25 in blue.

model depicts a planar conformation, molecular dynamics simulations suggest that two IP6 molecules can also be accommodated when they adopt non-planar orientations [14].

To investigate the importance of residue K25 in HIV infection we mutated it to either arginine, alanine or asparagine. Viral production of the K25 mutants was reduced by less than 2-fold compared to WT (S2A and S2B Fig) and no severe defect of Gag-processing could be determined either in virions (S2C Fig) or in cells (S2D Fig). However, the K25A and K25N mutants showed a profound loss of infectivity, with a reduction relative to wild-type of > 3 logs (Fig 2A and 2B). This loss of infectivity is very similar to that displayed by the IP6- and nucleotide-binding mutant R18G. Mutant K25R was substantially more infectious than K25A, with a ~ 1 log reduction in infection, highlighting the importance of maintaining a positive charge at this position. It showed a similar decrease in infection as P38A, a known CA instability mutant [15, 16].

**Table 1. Data collection and refinement statistics.** The CA-IP5 hexamer complex was deposited in the PDB with code 6R6Q. Statistics for both data integration and model refinement are given.

| | 6R6Q |
|---|---|
| **Data collection** | |
| Space group | P6 |
| Cell dimensions | |
| $a, b, c$ (Å) | 90.69, 90.69, 56.96 |
| $\alpha, \beta, \gamma$ (°) | 90.0, 90.0, 120.0 |
| Resolution (Å) | 78.55–2.73 (2.79–2.73) |
| $R_{\mathrm{meas}}$ | 15.2 (63.0) |
| $CC_{1/2}$ (%) | 99.7 (87.4) |
| $I / \sigma I$ | 10.2 (3.2) |
| Completeness (%) | 92.9 (92.9) |
| Redundancy | 6.5 (6.8) |
| Resolution (Å) | 2.7 |
| No. reflections | 7234 |
| $R_{\mathrm{work}} / R_{\mathrm{free}}$ | 0.22/0.28 |
| No. atoms | 1676 |
| Protein | 1612 |
| Ligand/ion | 64 |
| Water | 0 |
| $B$-factors | |
| Protein | 37.0 |
| Ligand/ion | 85.6 |
| Water | 0.00 |
| R.m.s. deviations | |
| Bond lengths (Å) | 0.02 |
| Bond angles (°) | 1.90 |

*Values in parentheses are for highest-resolution shell.

A similar charge-dependent pattern of behaviour is seen when mutating R18, with R18K being more infectious than R18G (Fig 2C). To assess functional independence of the two IP binding sites, R18 and K25, we compared the infectivity of each single conservative mutation to a double-mutant. Mutant R18K/K25R was substantially less infectious than either single mutant alone, suggesting that mutation while preserving charge at one site does not prevent function at the other (Fig 2C).

Previously it has been shown that there is a dose-dependent requirement for arginine at position 18 and that the loss in infectivity upon mutation correlates closely with loss of DNA synthesis[2]. To test whether the reduction in infectivity upon K25 mutation is similarly associated with a failure to undergo reverse transcription, we made a series of chimeric viruses by transfecting cells with different ratios of wild-type and K25A Gag plasmids. Previous work has shown that there is a strong correlation between input plasmid ratios and the ratios of the wild-type and mutant CA proteins in viral particles [17]. We observed a similar dose-dependent decrease in the infectivity of K25A chimeric viral particles (Fig 2D) as described for R18G [2]. Infection was broadly maintained in viruses predicted to have at least 4 of 6 lysines at position 25 but substantially reduced in mutants with < 3. This is similar to previous data with R18G and provides further evidence that produced viruses are chimeric, as a mixture of fully wild type and fully mutant viruses would give rise to a linear relationship. Importantly, this

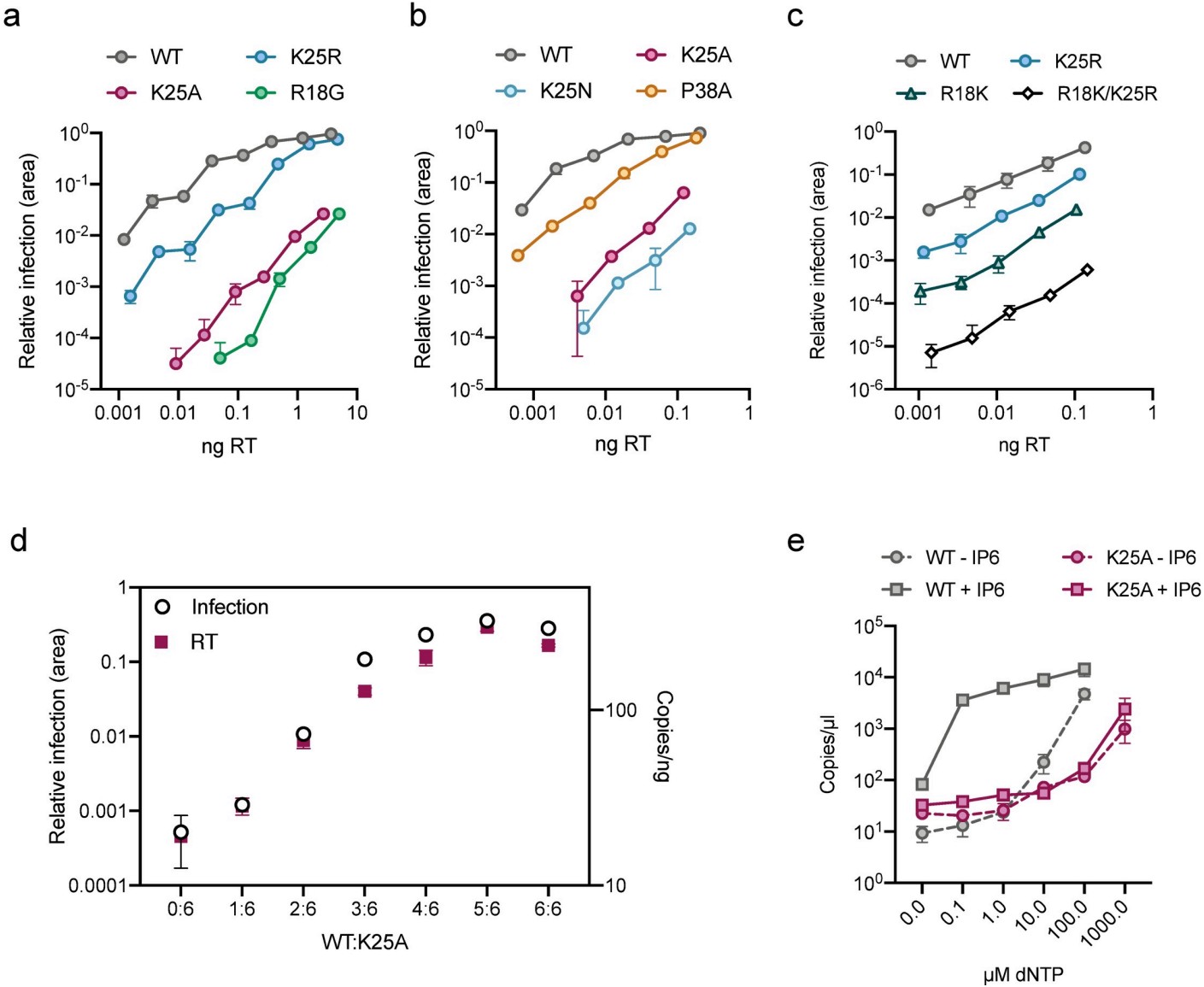

**Fig 2. K25 is essential for reverse transcription and infection.** (a&b&c) Infectivity of WT HIV and selected mutants. Infectivity is measured using an IncuCyte and determined as the proportion of cell area that is GFP +ve at a given viral dose (in ng RT). (d) Matching infectivity (GFP +ve cell area) and reverse transcription (strong-stop, RU5, at 4 hours post-infection) of chimeric viruses produced with an increasing ratio of WT:K25A Gag. (e) ERT assay measuring the synthesis of strong-stop DNA in the presence of DNase and at increasing concentrations of dNTPs. WT and K25A cores with and without IP6 (50 μM). Data is average of three biological replicas and has been normalized to the copies of RU5 measured in the absence of dNTPs. Error bars in infection experiments depict mean ± SD of three replicates from one experiment representative of three independent experiments.

pattern of infection loss was closely paralleled with a reduction in DNA synthesis. Thus, mutation of K25, like R18, results in viruses that cannot carry out reverse transcription. The data on chimeric R18 viruses was previously interpreted as indicating a role for R18 in importing nucleotides for encapsidated reverse transcription (ERT). To directly test whether mutating K25 impacts on the efficiency of DNA synthesis, we carried out a series of ERT experiments comparing in vitro reverse transcription of isolated viral cores (Fig 2E). To ensure that WT and K25A cores have incorporated similar levels of reverse transcriptase we measured enzymatic activity by RT ELISA (S2E Fig). ERT experiments were then carried out in the presence

of DNase to ensure that only DNA that is synthesised and protected within cores is measured. We observed an increase in DNA synthesis in wild-type cores upon titration of increasing concentrations of dNTPs (Fig 2E). In contrast, mutant K25A was far less efficient at carrying out reverse transcription. At a concentration of 100 μM dNTPs, where DNA synthesis in WT cores had increased > 2 logs, K25A showed very little activity. Importantly, while K25A is capable of carrying out reverse transcription and shows activity that is proportional to dNTP dose, it requires a > 10-fold higher concentration of dNTPs to achieve comparable levels of DNA synthesis as WT (WT gives > $10^3$ copies at 100 μM dNTP whereas K25A requires 1000 μM for slightly fewer copies). This result would be consistent with mutation of K25A reducing the efficiency of dNTP import. However, we have previously shown that increasing dNTP concentrations can promote the accumulation of synthesised DNA indirectly by stabilising the isolated viral cores. To test whether K25A undergoes reduced reverse transcription because of reduced stability or dNTP import we repeated our ERT experiments in the presence of IP6. When IP6 was added to WT cores, maximal DNA synthesis was observed at significantly lower dNTP concentrations, consistent with a stabilisation effect. In contrast, addition of IP6 had no effect on K25A suggesting that a defect in either dNTP import and/or ability of IP6 to stabilise the capsid is limiting DNA synthesis in this mutant (Fig 2E).

A limitation of the ERT assay is that while it is impacted by core integrity, it does not directly measure stability. Thus, it is possible that K25A did not respond to IP6 addition because the mutant no longer binds or is stabilised by the ligand. To assess whether K25 alters the ability of IP6 to stabilise the HIV capsid, we measured the thermal stability of disulphide-linked capsid hexamers in the presence or absence of phosphate ligands. Importantly, we found that K25 mutation did not prevent the binding of dATP or IP6 and a similar degree of stabilisation was observed as with wild-type protein (Fig 3A). Binding was also observed for IP5 and ATP (S3 Fig). In contrast, mutation of R18G leads to a loss of the ability to bind ligands (Fig 3A). Moreover, the affinity of Atto488-labelled ATP binding to pre-assembled hexamers was not affected by K25A (Fig 3B). These observations suggest that, within pre-assembled hexamers, K25 does not significantly contribute to IP6-mediated stabilisation or that this is not detectable in pre-stabilised hexamers. IP6 is incorporated into budding virus and is required for the formation of mature, infectious virions [3]. To assess whether K25 has a role in the assembly of mature capsids we examined mutant and wild-type virions by electron cryotomography (cryoET). Whereas the majority of wild-type virions contained mature cores, no complete cores were observed for K25A (Figs 3C and S4 and S5). This was not due to a failure in immature lattice processing because a similar number of immature cores were observed in WT and K25A virions and CA processing was identical (Figs 3C and S2C). These data suggest that the poor infectivity of K25A is likely to be caused, at least in part, by inefficient mature assembly. Given that K25A infectivity is ~1000-fold lower than WT, the proportion of correctly formed K25A mature cores could be as low as 0.1%. This may explain why we were unable to observe any K25A capsids; it would require the sampling of thousands of tomograms.

While cryoET provides insight into the efficiency of maturation and capsid morphology, it does not reveal whether assembled cores are stable. To assess capsid stability, we carried out a series of TRIM5 abrogation assays by infecting cells expressing rhesus macaque TRIM5. TRIM5 restricts HIV infection by binding directly to the capsid and assembling a hexagonal lattice around the virus[18]. Partly due to this mechanism of action, TRIM5 activity becomes saturated at a high multiplicity of infection. We co-infected cells with a consistent dose of WT virus that encodes a GFP reporter gene and a variety of mutant viruses that encode an RFP reporter gene (for instance, a single infection includes both WT-GFP and K25R-RFP). We measured both the RFP and GFP signal as we added increasing doses of RFP virus (Fig 3D).

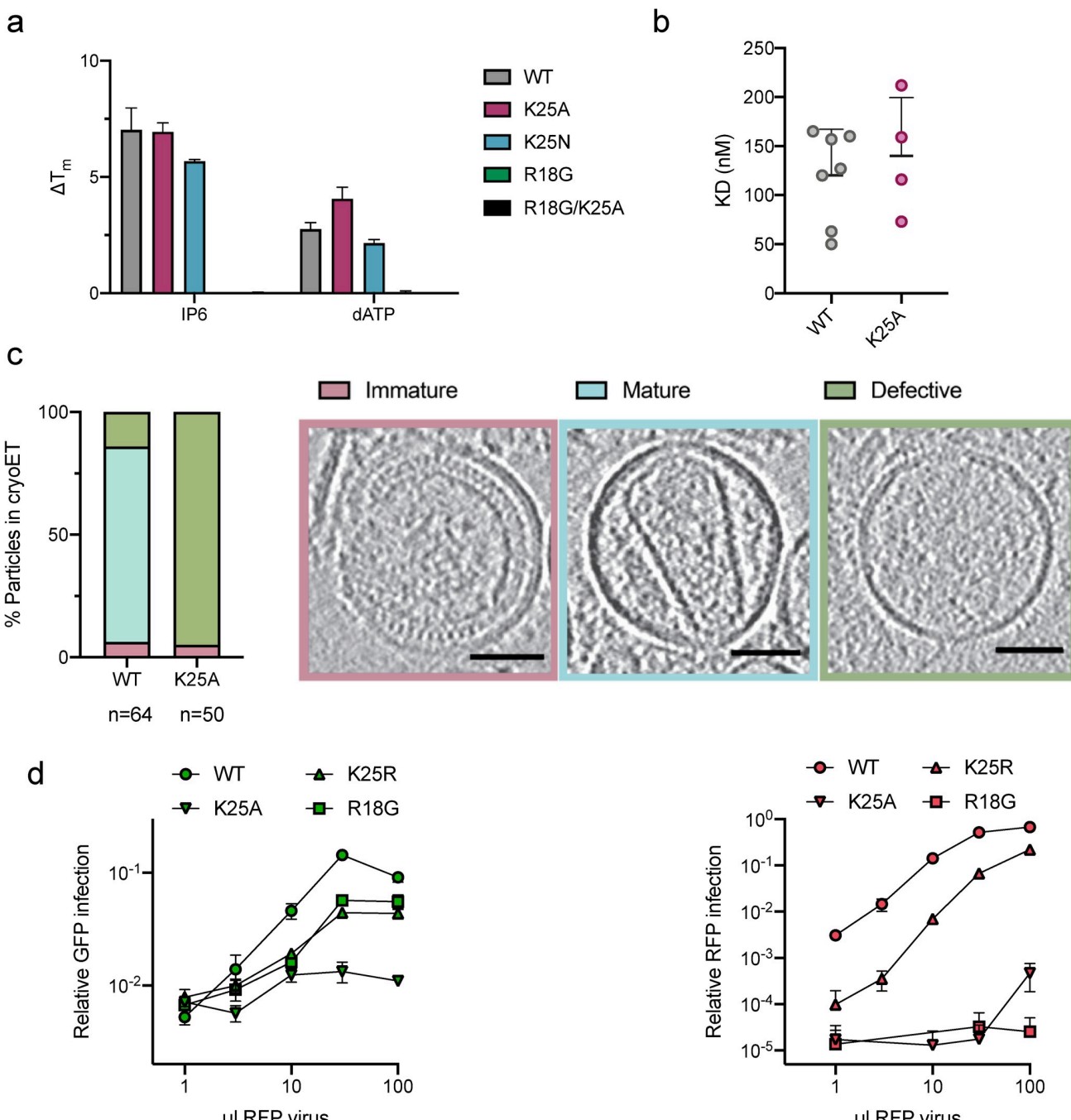

**Fig 3. K25 is required for the production of HIV virions with mature capsid cores.** (a) Changes of NanoDSF-derived melting temperature ($\Delta$Tm's) of K25A, K25N, R18G and R18G/K25A in the presence of different polyanions compared to the respective disulfide-stabilised hexamers in absence of ligands. Error bars depict mean ± SD of technical replicates. (b) Dissociation constants ($K_D$'s) of Atto488-ATP binding to self-assembled CA A204C/A92E cones determined by TIRF microscopy. (c) Cryo-ET analysis of infectious WT or K25A HIV virions. Virions were classified as either immature, mature or defective and slices through representative tomograms of each are shown. Scale bars, 50 nm. A total of 64 WT and 50 K25A viral particles were analysed and the frequency of each phenotype is plotted as a percentage. Immature viral particles are labelled in pink, mature in blue, and empty in green. (d) TRIM5 abrogation assay. Cells expressing Rhesus TRIM5 were infected with a consistent dose of WT GFP-expressing virus and increasing doses of RFP-expressing WT, K25A and K25R viruses. The level of infection of each virus (green symbols for GFP, red symbols for RFP) was determined as the proportion of cell area that was GFP or RFP positive. Increasing doses of WT RFP virus results in an increase in RFP positive cells but also in GFP positive cells, because TRIM5 becomes saturated with RFP virus and no longer restricts infection by the GFP virus. Viruses that do not contain capsids capable of recruiting and maintaining TRIM5 binding will not saturate. Error bars depict mean ± SD of three replicates from one experiment representative of two independent experiments.

When increasing doses of WT RFP virus were added, there was a corresponding increase in GFP signal from the co-infecting WT GFP virus, despite there being no change in the amount of GFP virus added (Fig 3D). This increase in GFP signal occurs because TRIM5 becomes saturated by the RFP virus and can no longer efficiently block GFP virus infection. However, we did not see the same phenomenon when using a K25A RFP virus. K25A RFP virus gave poor RFP expression and there was no increase in GFP expression from the co-infecting WT GFP virus. This indicates that not only is K25A poorly infectious but it also has too few stable capsids to efficiently saturate TRIM5. We repeated this experiment with K25R. This time the K25R RFP virus was able to dose-dependently saturate TRIM5 and an increase in the GFP signal was observed. The saturation ability of K25R RFP virus was ~ 1 log less efficient than WT virus and closely correlates with its 1 log less infectivity (Fig 2A). This suggests that K25R virus is less infectious because it has fewer stable capsids and not because it is less capable of mediating reverse transcription. R18G-RFP virus had a similar ability to saturate TRIM5 as K25R, but R18G-RFP infectivity was drastically lower compared to K25R (Fig 3D, RFP virus). This suggests R18G viruses form higher order capsid assemblies that are largely non-infectious. This is consistent with previous data showing that R18G can form cores but is unable to reverse transcribe in cells [2, 10, 19].

As the TRIM5 abrogation data suggested that K25 mutant viruses might be less stable, we decided to investigate this in more detail using single-molecule TIRF microscopy. We used a similar experimental approach as described previously[20, 21], in which virions are captured then permeabilised by a pore-forming toxin (Fig 4A). This enables fluorescent-labelled CypA in bulk solution to pass into the virion through the membrane pore to bind and 'paint' any intact capsid lattices that are present. The lifetime of each capsid can then be measured by monitoring the persistence of the fluorescence signal at all locations corresponding to viral particles through time (Fig 4B). We then used survival analysis of the capsid lifetimes to obtain estimates of capsid stability (Fig 4C). Using this approach, we compared the behaviour of WT and K25A viruses in the presence and absence of IP6. As previous, we observed that WT capsids have an intrinsic half-life of ~5–10 minutes, but this can be greatly extended for most capsids through the addition of IP6 (Fig 4C). In contrast, with K25A the CypA paint signal is extremely short-lived for most virions and the survival curve approaches baseline with a half-life of less than 1 minute. Moreover, addition of either 0.1 or 1.0 mM IP6 failed to confer any stability on K25A. This was also true for K25R: while K25R was slightly more stable than K25A in the presence of IP6, this effect is negligible when compared to WT (Fig 4C). While these results could indicate that K25 mutants cannot be stabilised by IP6, it is more likely that there are too few properly formed and IP6-responsive capsids present in the population to be able to detect in the assay. This is supported by calculating the fraction of capsids that remain intact 20 minutes after permeabilization. A fraction of WT capsids (on average ~20%) remain at this time point and indeed the majority (on average ~60%) are intact in the presence of IP6 (Fig 4D). In contrast, only 2–3% K25 mutant capsids remain after 20 minutes irrespective of whether or not IP6 is present. These data indicate that the fraction of stable (or IP6-responsive) capsids for K25 mutants is reduced by at least 10-fold compared to wild-type, but whether there are stability differences between K25A and K25R cannot be determined with this method. Monitoring capsids via CypA paint also allows a relative assessment of their size, because the fluorescence intensity is proportional to the number of CA molecules in the lattice. Consistent with K25 mutant virions having an assembly defect, the distribution of lattice sizes of the mutants is shifted to significantly smaller values than wild-type (Fig 4E).

The above data suggest that K25 plays an important role in capsid assembly and virus infectivity. However, whether K25 mediates its effects via polyanion binding is unclear. The IP5-complexed structure suggests K25 coordinates a second inositol phosphate but neither IP5

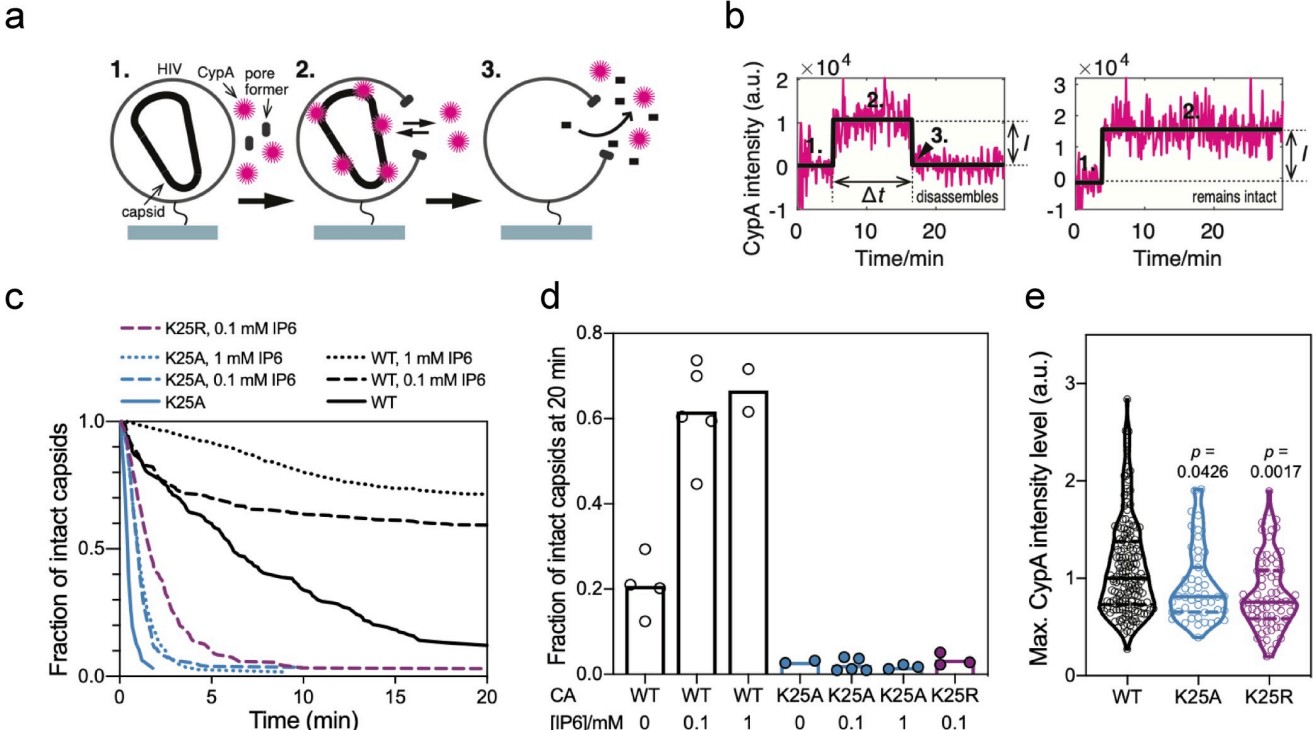

**Fig 4. CA K25 mutants capsids are short-lived in the absence and presence of IP6.** (a) Schematic of the capsid uncoating assay using TIRF microscopy and the CypA paint method. Immobilized viral particles are permeabilized in the presence of fluorescently labeled CypA while recording fluorescence traces at the locations of single virions (1). CypA binds to the capsid, resulting in the appearance of a stable fluorescence signal (2). Capsid disassembly results in the disappearance of the CypA signal (3). (b) Example traces for a capsid that disassembles (left) and a capsid that remains intact (right) during the acquisition time. The fluorescence traces (magenta) are analysed by step fitting (black) to extract lifetime ($\Delta t$) and intensity ($I$) for each capsid. (c) Capsid survival curves generated from single-virion uncoating traces. CA K25A mutant capsids fall apart rapidly and IP6 does not stabilise CA K25A or K25R capsids, unlike wild type capsids. (d) Bar graph of the fraction of capsids that remain intact 20 min after permeabilization of the viral membrane. (e) Intensity distributions of the CypA paint signal. The intensity is proportional to the size of the CA lattice. The lower median intensity for K25 mutants compared to wild type suggests the presence of a smaller CA lattice. Comparisons using the Kolmogorov-Smirnov test.

nor IP6-stabilisation of hexamers is affected by its mutation, in contrast to removal of R18. K25 is in proximity of several other charged residues that are present on the same helix–E28, E29 and K30. These residues could potentially participate in hydrogen bond or electrostatic interactions across monomers within a hexamer; K25 with E29 and K30 with E28 (Fig 5A). Using our IP5 structure, we also modelled the arrangement within a pentamer based on an EM structure of HIV-1 capsid pentamers in intact virions (5MCY[22]) (Fig 5B). In this case, only K30 is positioned for inter-subunit interaction. This is broadly similar to a previous assessment carried out on capsid models 3J3Q and 3J3Y [23], which found that E28-K30 hydrogen bonds were predicted in 24% of hexamers and 86% of pentamers whereas E29-K25 hydrogen bonds were suggested in 20% of hexamers and <40% of pentamers [24]. We therefore considered the possibility that K25 may participate in capsid assembly not through coordination of IP6 but as part of a charged interaction network and specifically through interaction with E29. To test this, we mutated each of the four charged residues to alanine and measured their impact on infection. Interestingly, while mutants K25A, E28A and K30A all had a profound infectivity defect, E29A behaved similarly to wild-type (Fig 5C). The similar phenotypes of E28A and K30A are consistent with the notion that they participate in a direct interaction that is important for capsid assembly, particularly pentamer formation. In contrast, the failure of E29A to phenocopy K25A suggests that interaction between these residues is not

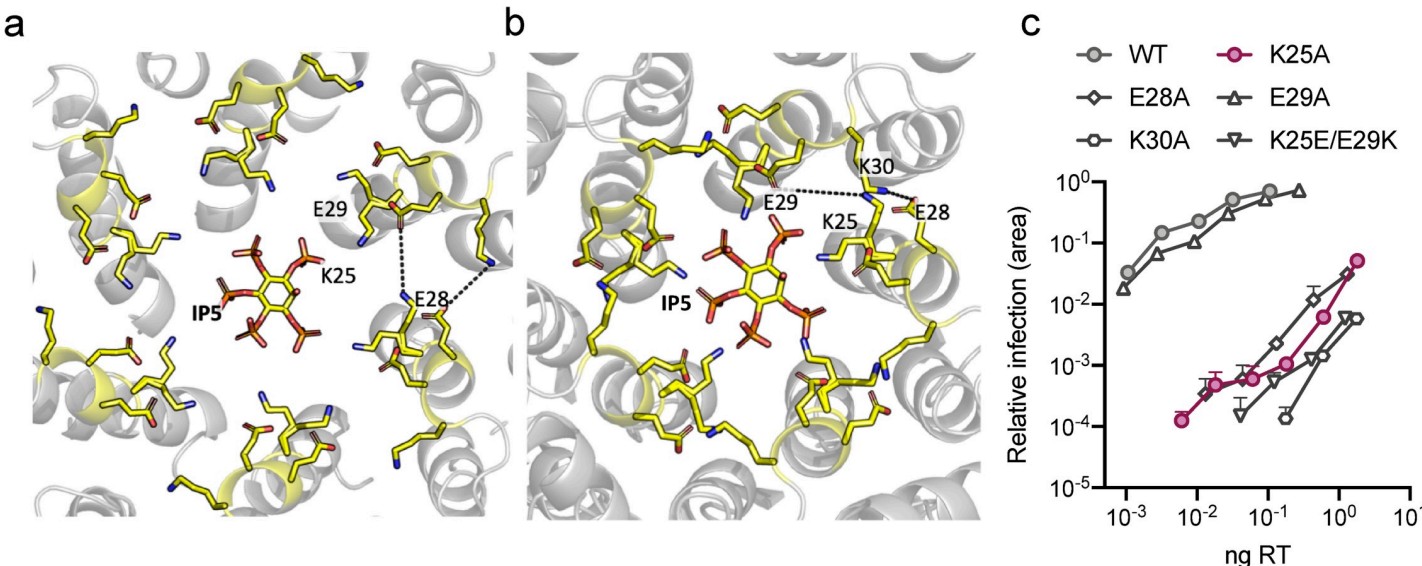

**Fig 5. K25 is not required to interact with E29 in hexamers or pentamers.** (a) IP5:CA hexamer structure showing the location of K25, E28, E29 and K30. Only the second IP5 molecule in proximity to K25 is depicted. Dotted lines indicate potential interactions. (b) Model of an IP5:CA pentamer, generated using monomers from the hexamer in (a) and the pentamer cryo-EM structure 5MCY[22]. (c) Infectivity of WT HIV and selected mutants. Infectivity is measured using an IncuCyte and determined as the proportion of cell area that is GFP +ve at a given viral dose (in ng RT). Error bars in infection experiments depict mean ± SD of three replicates from one experiment representative of three independent experiments.

important for viral fitness. To confirm this result we performed a charge swap experiment that would maintain any interaction but switch the respective positions of the residues within the hexamer or pentamer pore. The double mutant K25E/E29K showed poor infectivity, consistent with the importance and location of K25 to allow IP6 binding rather than E29K interaction (Fig 5C).

We hypothesised that K25 uses IP6 to increase the efficiency of mature capsid assembly, in addition to any role in nucleotide import. We therefore sought a way to determine if it is the specific loss of IP6 coordination that reduces assembly or whether K25 is simply unstable. We carried out a series of *in vitro* assembly reactions using recombinant CA protein. We first incubated WT, K25A, K25R, K25N, P38A and R18G viruses under high salt conditions previously shown to induce the assembly of capsid tubes [25]. We monitored the kinetics of assembly by measuring the absorbance at 350 nm and collected samples at the end of each experiment for analysis by negative stain electron microscopy (Fig 6A–6C). We observed that K25A, K25R and K25N mutant viruses were capable of assembling capsid tubes like WT (Fig 6A–6C). Moreover, their assembly kinetics were similar, albeit with an initial lag for K25A. These data indicate that mutation of K25 to alanine or arginine does not intrinsically alter the CA such that it is incapable of higher order assembly. As previously described, P38A showed a slightly faster tube assembly compared to WT [15]. In contrast, R18G did not form tubes but small spheres [26]. Next, we carried out WT assembly experiments in low salt conditions and at a series of different IP6 concentrations (S6A Fig). We observed that without either high salt or IP6 no assembly takes place, as assessed by absorbance at 350 nm. However, as we titrated increasing concentrations of IP6 we observed a striking dose-dependent increase in both the kinetics of assembly and the quantity of assembled material (S6A Fig). Choosing the highest dose of IP6, we repeated these experiments and compared WT CA to mutants K25A and K25R. We observed a complete loss of assembly for K25A and K25N and a decreased assembly for K25R (Fig 6D). Analysis by negative stain revealed that under IP6 conditions, the WT CA

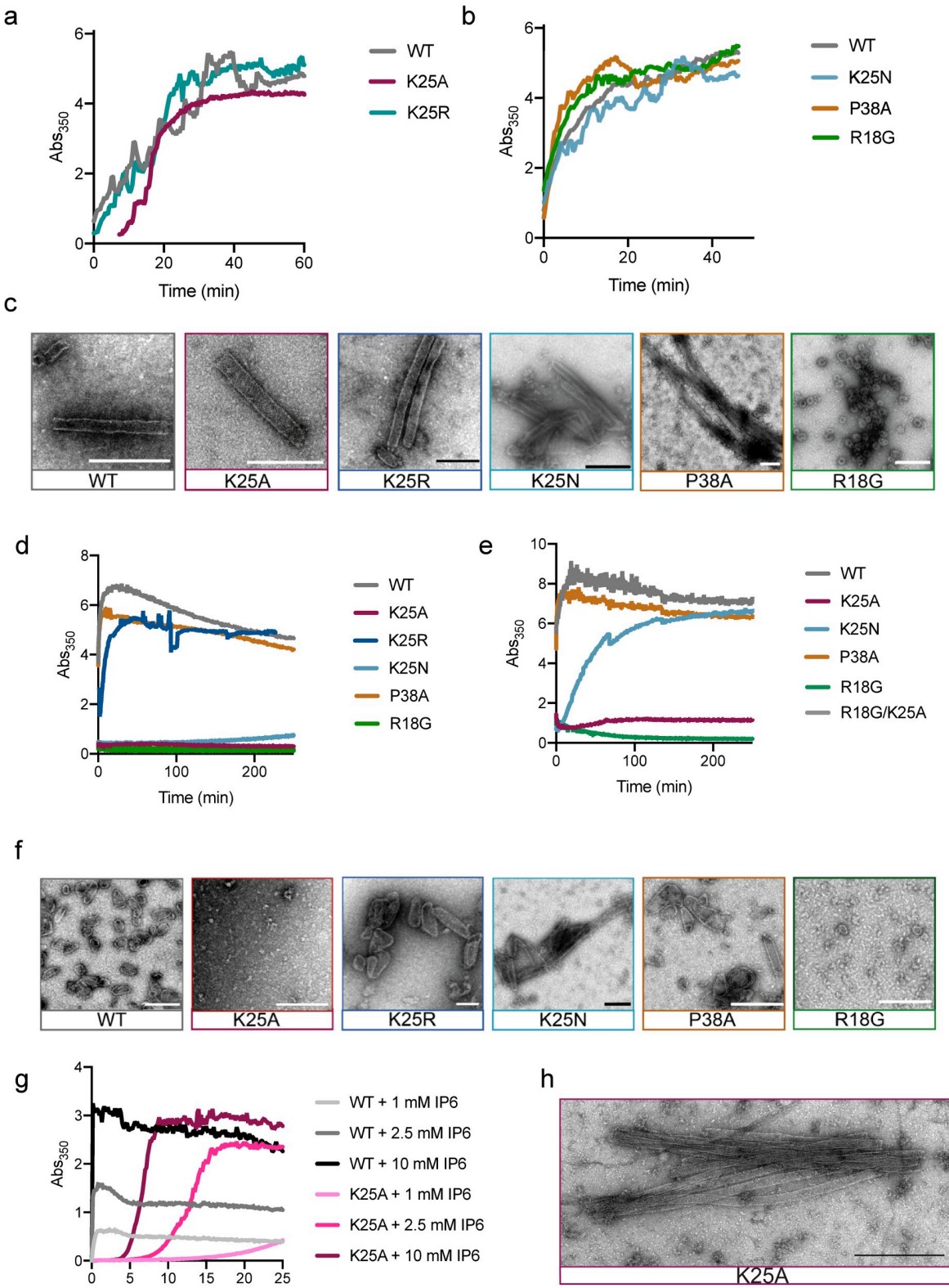

**Fig 6. K25 is required for IP6-mediated assembly of HIV CA.** In vitro assembly reactions were monitored in real-time be measuring the absorbance at 350 nm. (a&b) Assembly reactions using 75 μM capsid protein and 2.5 M NaCl in 50 mM MES pH 6.0 (c) Negative stain EM images of material taken from (a&b). Scalebar: 200 nm. (d) Assembly reactions using 100 μM WT and mutant capsid protein

with 1.25 mM IP6 and (e) 200 μM WT and mutant capsid protein with 6 mM IP6 in 50 mM MES pH 6.0, 40 mM NaCl (f) Negative stain EM images of material taken from (d&e). Scalebar: 200 nm (g) Assembly reactions of 250 μM WT or K25A CA in 50 mM MES pH 6, 100 mM NaCl and a range of IP6 concentrations up to 10 mM. (h) Negative stain EM images of material taken from (f).

forms conical capsids reminiscent of mature HIV cores, as previously reported (Fig 6F). K25R was also able to form mature capsids, albeit with reduced yield and an increased proportion of larger cores. In contrast, neither capsid cores nor tubes were observed for K25A and K25N. As expected, R18G was not able to assemble. The hypostability mutant P38A, however, assembled cores with similar kinetics as WT.

As K25A and K25N tubes had been observed under high salt assembly conditions, we varied CA and IP6 concentration and measured 350 nm absorbance (Fig 6E). At increased capsid (200μM) and increased IP6 (6 mM) concentrations we observed K25N assembly to tubes and cores (Fig 6E and 6F). K25A only showed a slight increase in absorbance compared to R18G and R18G/K25A which were not able to assemble at all.

Thus, we measured the K25A assembly reaction for 24h. At 1 mM IP6 concentration, the WT CA reaction was complete within 1 hour (Fig 6G). For K25A, an increase in absorbance began to be observed after 15 hours. Increasing the IP6 concentration led to a dose dependent increase in both the kinetics and turbidity of both WT and K25A reactions. At 10 mM IP6, the WT reaction was complete within 10 minutes, while the K25A reaction reached plateau after ~ 10 hours. To determine the morphology of the assembled K25A CA, we analysed the sample by negative stain electron microscopy and observed tubes with dimensions of 44.7 ± 5.7 nm, consistent with that observed for WT and mutant CA under high salt conditions (Fig 6H). Since these are very high IP6 concentrations we set up an assembly reaction with 500 μM WT and K25N with 10 μM IP6 incubated them for several hours, and analysed them via EM pictures (S6B Fig). Taken together, these results suggest that K25A mutants are severely impaired for IP6-driven assembly of capsid tubes, while capsid cones do not form or do so too rarely to be detected by in vitro methods. Given that tubes are known to be comprised of a hexamer lattice whereas cones require pentamers, this suggests that K25:IP6 interaction may be particularly crucial for pentamer formation. This is consistent with the closer packing of K25 side chains predicted in the pentamer. We therefore measured the thermal stability of disulphide-linked capsid pentamers with IP6 and observed an increase of 4°C in presence of IP6 (S6C Fig). This suggests the IP6 can bind to pentamers and stabilise them.

## Discussion

The mature HIV capsid contains positively charged pores capable of binding multiple polyanions, including IP6 and dNTPs[1–4]. IP6 promotes the assembly and stability of mature capsids, while dNTP import through the pore provides a mechanism to allow DNA synthesis inside an otherwise impermeable protein lattice. The relative importance of these roles and how they have driven pore evolution is unclear. In this study, we sought to investigate whether a positively charged ring within the pore, created by K25, is involved in polyanion binding and how it impacts HIV infectivity. Taking advantage of the increased planarity of IP5 over IP6, we could demonstrate that two polyanions can bind the pore simultaneously. One molecule binds above the R18, while a second molecule binds above the K25 ring. Mutating K25 results in a dramatic loss in infectivity. An attractive explanation for the requirement of K25 within the pore is that it ensures directional import of nucleotides that are attracted by initial R18 binding. However, molecular dynamics simulations that model nucleotide movement through the pore have provided conflicting results[12, 13]. Mutation of K25 does lead to loss of reverse transcription both in cells and in vitro inside purified capsids, consistent with a role in import.

Cryo-tomography on purified HIV virions shows that K25A mutants also have a maturation defect; indeed we were unable to find convincing examples of properly mature capsids. This suggests that inefficient mature capsid formation contributes to the reduced infectivity of K25A. Single molecule TIRF studies confirmed a structural defect; K25A capsid lattices were quickly lost upon virion permeabilization and addition of IP6 failed to measurably rescue stability. Finally, *in vitro* assembly experiments demonstrated that K25 uses IP6 to promote capsid assembly. Mutants K25A, K25R and K25N were capable of forming capsid tubes in the presence of high salt, a widely-used condition for promoting assembly[25], and with similar kinetics and morphology as WT. However, the ability of IP6 to promote the assembly of capsid cones in the absence of high salt[1] was lost in K25A. Addition of IP6 did allow K25R and K25N to form cones, albeit with markedly reduced kinetics and with an increased proportion of larger cones, or tubes in case of K25N. No loss of IP6-mediated stabilisation was observed upon mutating K25 in NanoDSF experiments, however these were performed using hexamers that are already stabilised by disulphide binds. Taken together, these data confirm that K25 is involved in the IP6-driven assembly of mature capsid cores and likely explains why no K25A mature capsids were observed in virions.

The native HIV capsid is very fragile with around 70% of capsid point mutations leading to virions which a have strongly decreased infectivity compared to WT virus [10]. Reduced or increased capsid stability was shown to have a strong impact on HIV-1 replication and reverse transcription [10, 16, 27]. We compared the behavior of K25A to P38A, a capsid instability mutant which was previously shown to have decreased infectivity and impaired reverse transcription [15, 16]. In our experiments, P38A had a similarly reduced infectivity as K25R, but it was able to form cores in both high salt and IP6 conditions. This suggests that K25 mutants (and R18) are distinct from other mutations affecting the capsid. Nevertheless, it is important to note that IP6 binding and hexamer stability are thermodynamically interdependent, because the IP6 binding site is only present in the hexamer and binding stabilises the hexamer form. Thus, we would predict that reducing hexamer stability will decrease IP6 binding, while decreasing IP6 binding will reduce hexamer stability. The degree to which different capsid mutants alter these properties, in terms of capsid assembly within the virion and capsid stability within the cell, will require careful investigation.

While these results highlight that K25 interacts with inositol phosphates and that this interaction is used to promote assembly, it is unclear exactly how this is achieved. It is tempting to speculate that K25:IP6 interactions are particularly important for stabilising pentamers and allowing cones, rather than just tubes, to form. We showed that IP6 can bind to disulphide-stabilised pentamers. But it has to be taken into consideration that the disulphide pentamer structure differs from cryo-ET pentamer structures [22]. However, a recent MD simulation using a cryo-ET pentamer structure confirms that IP6 can (theoretically) bind and stabilise pentamers [14]. Packing within the pore is tighter in pentamers than hexamers. Our models suggest that K25 forms a ring in which the side-chains are 8 Å closer in pentamers than hexamers. Thus, electrostatic repulsion is likely to be more of a destabilising force in pentamer formation. If K25 were only a destabilising force then K25A should more efficiently form pentamers whereas the fact that it assembles tubes but not cones in vitro, suggests that K25 is participating in an interaction that actively stabilises pentamers.

While the details of how K25 participates in assembly are of interest, perhaps the more important question is why HIV has evolved this particular mechanism. Mutant K25N has been described as allowing similar levels of mature capsid formation as WT[13], suggesting that IP6-stabilisation at this position, whether in hexamers or pentamers, is not a requirement for assembly. We favour the hypothesis that K25 is maintained by HIV because it is part of the nucleotide import mechanism that allows encapsidated reverse transcription to take place.

IP6:K25 interaction may therefore be required for capsid assembly only as a consequence of K25 being essential for dNTP import. If assembly were the only criteria then this interaction could be lost, albeit with the limitation that it be replaced by some form of packing interaction (asparagine may be possible but alanine is not). Further work will be required to test whether K25 is evolutionarily conserved by HIV because of its role in reverse transcription and, if possible, dissect this from its impact on capsid assembly and stability.

## Materials & methods

### Cells and plasmids

293T CRL-3216 cells were purchased from ATCC. All cells are regularly tested and are mycoplasma free. HEK293T and HeLa cell lines were cultured in Dulbecco's modified Eagle's medium (DMEM) with 10% FBS, 2 mM L-glutamine, 100 U/ml penicillin, and 100 mg/ml streptomycin (GIBCO) at 37˚C with 5% $CO_2$). Replication deficient VSV-G pseudotyped HIV-1 virions were produced in HEK293T cells using the lentiviral packaging plasmid pMDG2, which encodes VSV-G envelope (Addgene plasmid # 12259) pCRV GagPol [28] and CSGW [29] as described previously [30]. Mutagenesis of CA was performed using the Quick-Change method (Stratagene) against pCRV-1 Gag-Pol.

### Infection experiments

For infection experiments with 293T, cells were seeded at $1x10^4$ cells per well into 96-well plates and left to adhere overnight. The media was replaced with FluoroBrite-DMEM (GIBCO) with 10% FBS, 2 mM L-glutamine, 100 U/ml penicillin, and 100 mg/ml streptomycin and 5 mg/ml polybrene. Indicated amounts of virus were added, and the plates were scanned every 8 h for up to 72 h in an IncuCyte (Satorius) to identify GFP-expressing cells. Infections of HeLa cells was performed in presence of 5 μg/ml polybrene in 6-well plates seeded with 10 5 cells per well. The plates were scanned at the indicated time points in an IncuCyte (Satorius) to identify GFP-expressing cells.

### HIV quantification

HIV genomes were quantified using the Cell-to-$C_T$ Kit (Invitrogen). 100 μl viral supernatant was centrifuged ions to pellet the viral particles, then resuspended with 20 μl lysis buffer with DNaseI 1/100 and incubated at room temp for 10 mins. 2 μl Stop solution was added to terminate the reaction. 2 μl Stop solution was added to terminate the reaction. 10μl RT Buffer (2x) was mixed with 0.5 μl RT enzyme mix and 9.5 μl lysate and and incubated at 37˚C 1 h, followed by 95˚C 5 min run on a ABI StepOnePlus Real Time PCR System (Life Technologies) (37˚C 1hr, 95˚C 5min). The qPCR was performed on a ABI StepOnePlus Real Time PCR System (Life Technologies) using 2 μl RT product, 5 μl TaqMan Fast Universal PCR Mix (ABI) and 0.5 μl GFP primer-probe (GFP) in a 10 μl reaction ((GFPF (CAACAGCCACAACGTCTAT ATCAT), GFPR (ATGTTGTGGCGGATCTTGAAG) and probe GFPP (FAM-CCGACAAGC AGAAGAACGGCATCAA-TAMRA). CSGW plasmid with calculated copies was used as a standard.

The level of RT enzyme was quantified using either a colorimetric RT assay kit (Roche) according to manufacturer's instructions or qRT-PCR as described previously with slight alterations [31]. In brief, 5 μl of viral supernatant was mixed with 5 μl lysis buffer (0.25% Triton X-100, 50 mM KCl, 100 mM Tris-HCl (pH 7.4), 40% glycerol) and 0.1 μl RNase Inhibitor and incubated for 10 min at room temperature before diluting to 100 with nuclease-free water. 2 μl of lysate was added to 5 μl TaqMan Fast Universal PCR Mix, 0.1 μl MS2 RNA, 0.05 μl RNase

Inhibitor and 0.5 μl MS2 primer mix, to a final volume of 10μl. The reaction was run on a ABI StepOnePlus Real Time PCR System (Life Technologies) with and additional reverse transcription step (42°C 20min) and followed by amplification.

## qPCR for in-cell reverse transcription products

Viral supernatants were treated with 250 U/ml DNase (Millipore) for 1 min to remove contaminating DNA. DNase treated viruses were added to cells as per infection protocol and incubated at 37°C. Cells were harvested at indicated time points and and the DNA was extracted using the DNeasy Blood and Tissue Kit (Qiagen) according to manufacturer's instructions.

Reverse transcription products were detected by qPCR from 2 μl DNA sample using TaqMan Fast Universal PCR Mix (ABI) and RU5 primers to detect strong-stop DNA40 (RU5 forward: 5' CTGGCTAACTAGGGAACCCA-3'; RU5 reverse: 5'-CTGACTAAAAGGGTCTG AGG-3'; and RU5 probe 5'-(FAM) TTAAGCCTCAATAAAGCTTGCCTTGAGTGC(TAM RA)−3') and GFP primers to detect first-strand transfer products (see above). Reverse transcription measurements are representative of 3 experiments with each point measured in triplicate. Results are represented as mean ± standard deviation.

## Preparation of HIV cores and encapsidated reverse transcription assay

Purification of HIV cores and the encapsidated reverse transcription assay were performed as described previously [4]. Briefly, supernatant containing VSV-G pseudotyped HIV-1 GFP was passed through a 0.45 μm nitrocellulose filter and pelleted through a 20% sucrose cushion (28,000 rpm at 4°C, 2h, Beckman SW32Ti rotor; Beckman Coulter Life Sciences). All following solutions were prepared in CPB (20 mM Tris (pH 7.4), 20 mM NaCl, 1 mM MgCl2). The pellets were resuspended in CPB and treated with DNase I for 2h (Sigma Aldrich) at room temp. A 80–30% sucrose gradient (5% steps) overlayed with 1% Triton X-100 in 15% sucrose. The resuspended virus was layered on top of the gradient and spun at 32,500 rpm at 4°C for 16hr with a Beckman SW40Ti rotor (Beckman Coulter Life Sciences). The gradient was fractionated, and the location of cores was determined via qPCR for RT (see above). Core-containing fractions were pooled, aliquoted and snap frozen before storage at −80°C. For the ERT assay, viral cores equalized (as determined by RT) by dilution in 60%. 10μl diluted cores were mixed with 100 mM Tris pH8, 100 mg/ml DNase I or BSA (as a negative control), dNTPs and or IP6 (were added at indicated concentrations) in a final reaction volume of 20μl concentrations. Reactions were incubated at room temperature for 16 hr. Reverse transcript products were detected using TaqMan Fast Universal PCR Mix (ABI) with RU5 primers to detect strong-stop DNA40 (described above).

## TRIM5 abrogation assay

Assay was performed essentially as described[2]. GFP or RFP VSV-G pseudotyped HIV-1 virus was produced as described above and concentrated by ultracentrifugation. Viruses were titrated on FRhK-4 cells at the indicated ratios and infection monitored by measuring GFP and RFP expression in cells using an IncuCyte (Sartorius).

## Protein production and purification

The CA proteins were expressed and purified as previously described [32, 33]. The disulfidestabilised CA hexamer purified as previously described with minor modifications [32, 33] Briefly, p24 hexamer protein was expressed in *E.coli* C41 cells, lysed and cleared by

centrifugation. The supernatant was precipitated in 25% ammonium-sulfate and the pelleted material was resuspended in 50 mM citric acid (pH 4.5), followed by dialysis against the same buffer with 20 mM 2-mercaptoethanol. Afterwards the protein was then dialysed into 50 mM Tris-HCl (pH 8.0), 1 M NaCl, 20 mM 2-mercaptoethanol. The reducing agent was removed by dialyzing against 50 mM Tris (pH 8.0), 1 M NaCl, and then finally into 20 mM Tris (pH 8.0), 40 mM NaCl. Reassembled hexamers were identified by non-reducing SDS–PAGE. All p24 capsid proteins constructs were further purified via anion-exchange columns followed by size exclusion chromatography (Superdex 200).

Plasmid pET11a bearing HIV-1 CA pentamer mutant (N21C/A22C/W184A/MA815A) was transformed into E. *coli* BL21 (DE3) competent cell. 1.5L LB media containing 100 μg/ml Ampicillin is inoculatsed with 15ml o/n culture and is induced with 0.5mM IPTG at $OD_{600}$ = 0.6 for 3h at 37˚C, 180rpm. The pentamer was purified like the hexamer (see above).

### Differential Scanning Fluorimetry

DSF measurements were performed using a Prometheus NT.48 (NanoTemper Technologies) over a temperature range of 20–95˚C using a ramp rate of 2.5˚C / min. CA hexamer samples were prepared at a final concentration of 200 μM monomer in PBS in the presence or absence of 4 mM DTT. dNTPs or competitors were added at 200 μM. DSF scans are single reads of three replicates and were performed at least three times.

### Turbidity assays

CA proteins were dialysed against 50mM MES (pH 6.0), 40 mM NaCl, 1mM DTT. CA proteins at a final concentration of 50–200 μM were mixed with NaCl (final concentration 2.5M) or IP6 (final concentration 200μM-10mM). The increase in $Abs_{350}$ was measured using a PHERAstar FSX Plate reader (BMG Labtech) in 384-well plate every 22 sec with shaking between each measurement.

### Negative stain

Samples from the turbidity assay were allowed to sediment overnight. 4 μl of sample was put onto a glow discharged carbon coated grid (Cu, 400 mesh, Electron Microscopy Services), washed and stained with 2% Uranyl-acetate. Micrographs were taken at room temperature on a Tencai Spirit (FEI) operated at an accelerated voltage of 120 keV and Gatan 2k × 2 k CCD camera. Images were collected with a total dose of ~30 $e^-/A^{\circ 2}$ and a defocus of 1–3 μm.

### Virus particle production for tomography

Virus-like particles were produced in HEK293T as described above. Supernatants were harvested and passed through a 0.45 μm filter followed by a 0.22-μm filter. The particles were concentrated by ultracentrifugation over a 20% (wt/vol) sucrose cushion (2 h at 28,000 rpm in a Beckman SW32 rotor; Beckman Coulter Life Sciences). Resuspended particles were applied to a 6–18% iodixanol gradient (1.2% increment steps) and centrifuged for 1.5 h at 250,000 × g in an Beckman SW40 rotor (Beckman Coulter Life Sciences) [34]. The virus containing fraction was diluted in 1:10 PBS und concentrated by ultracentrifugation (45 min,38,500 rpm in a Beckman SW40 rotor, Beckman Coulter Life Sciences). The pellet was resuspended in PBS and incubated at 4˚C overnight to allow full resuspension.

## Cryo-tomography

10-nm-diameter colloidal gold beads were added to the purified HIV-1 mutants. 4 μl sample-gold suspension was applied to a glow discharged C-Flat 2/2 3C (20 mA, 40 s). Grids were blotted and plunge-frozen in liquid ethane with a FEI Vitrobot Mark II at 15°C and 100% humidity. Tomographic tilt series were acquired between −40° and +40° with increments of 3°, on a TF2 Tecnai F20 transmission electron microscope equipped with a Falcon III Direct Electron detector at 200 kV using Serial-EM [35] under low-dose conditions at a magnification of 50000x and a defocus between -3 μm and -6 μm. The IMOD package was used to generate tomograms [36]. Alignment of 2D projection images of the tilt series was carried out using gold particles as fiducial markers. A 3D reconstruction was generated using back projection of the tilt-series.

## Crystallization, structure solution and analysis

CA hexamer protein was prepared exactly as described previously[4]. Crystals were grown at 17°C by sitting-drop vapour diffusion in which 100 nl protein was mixed with 100 nl precipitant and suspended above 80 μl precipitant in the MORPHEUS I screen [37].The structure was obtained from 12 mg/ml hexamer mixed with 1mM of myo-IP$_5$ and cryoprotected with precipitant supplemented with 20% MPD. Crystals were flash-cooled in liquid nitrogen and data collected at beamline I24 at Diamond Light Source. The data sets were processed using the CCP4 Program suite[38]. Data were indexed and integrated with iMOSFLM and scaled and merged with AIMLESS[39]. Structures were solved by molecular replacement using the model 6ES8 in PHASER[40] and refined using REFMAC5[41]. Between rounds of refinement, the model was manually checked and corrected against the corresponding electron-density maps in COOT[42]. Final figures were rendered in The PyMOL Molecular Graphics System, Version 1.5.0.4 Schrödinger, LLC. The model and data were deposited in the PDB database with code 6R6Q.

## TIRF biosensor affinity measurements

The affinity of Atto488-ATP (NU-805-488, Jena Bioscience) binding to CA lattices was measured using a capsid biosensor based on TIRF microscopy[43, 44]. Briefly, CA structures were self-assembled using a mixture of CA A204C/A92E (for cross-linking and increased solubility, respectively) and AF647-labelled CA K158C and then captured on the surface of a coverslip. Binding of Atto488-ATP (10–500 nM) to surface-immobilized fluorescent CA assemblies was then imaged by TIRF microscopy and quantified to obtain an equilibrium binding curve. The K$_D$ of the interaction was obtained by fitting the curve with an equilibrium binding model.

## Virus production for TIRF microscopy

Replication deficient HIV-1 virions without envelope protein were produced in HEK293T cells using pCRV-1 GagPol and pCSGW, biotinylated using EZ-Link Sulfo-NHS-LC-LC-Biotin (Thermo Scientific, 21338) and purified as described [20, 21].

## TIRF imaging of capsids in permeabilized viral particles

TIRF microscopy was carried out following the published method of Marquez et al[20, 21]. Briefly, biotinylated viral particles were captured onto coverslips attached to microfluidic flow cells and imaged using a custom built TIRF microscope with an ASI-RAMM frame (Applied Scientific Instrumentation), a Nikon 100 x CFI Apochromat TIRF (1.49 NA) oil immersion objective and NicoLase laser system. Immobilised virions were treated with imaging buffer

containing 200 nM PFO, to permeabilize the lipid envelope, and Alexa Fluor 568-labelled CypA (0.5–1 μM), to detect the capsid. TIRF images were then acquired with a frequency of 1 frame/6 s using a 561 nm laser with a 20 ms exposure time for excitation and an Andor iXon 888 EMCCD camera for detection. Single-virion fluorescence traces were extracted from the TIRF image stacks using the JIM Immobilized Microscopy analysis package (https://github. com/lilbutsa/JIM-Immobilized-Microscopy-Suite) and further analysed in MATLAB (The MathWorks, Inc) using software adapted from previous work [45]. Briefly, the duration and intensity of the CypA signal was extracted from fluorescence traces by step-fitting using change point analysis. The lifetime of each capsid was determined as the time difference between acquisition of Alexa Fluor 568-CypA upon permeabilization and loss of fluorescence upon capsid uncoating.

## Supporting information

**S1 Fig. Model of IP6 bound to HIV CA hexamer.** (a) Structural model of HIV hexamer with two IP6s using a hexamer structure with one IP6 (6ES8) and our hexamer IP5 structure. View showing the two IP6 molecules, one binding above R18 and one above K25. The axial phosphates of both molecules can be accommodated without steric hindrance. (b) View showing our IP5 hexamer structure.
(EPS)

**S2 Fig. Effect of K25 mutation on Gag-processing and virus release.** (a) RT assay of viral supernatant. Error bars depict mean ± SD of three replicates from one experiment representative of three independent preps. (b) Fold reduction of RT present in viral supernatant of mutants compared to WT in 2 biological replicates. (c) Western-blot of purified virions showing that WT and mutants have similar levels of p24 processing, consistent with normal maturation. (d) Western-blot of infected cells showing that WT and mutants have similar levels of p24. (d) RT assay to show similar incorporation of the enzyme in WT and K25A cores for ERT experiments. Error bars depict mean ± SD of three replicates from one experiment representative of three independent core preps.
(EPS)

**S3 Fig. K25A hexamers can bind polyanions.** NanoDSF-derived melting temperatures (Tm's) of WT or K25A disulfide-stabilised hexamers ± DTT and in the presence of different polyanions. Error bars depict mean ± SD of at least three technical replicates.
(EPS)

**S4 Fig. Gallery of selected Tomograms for WT HIV virions.** Slices through individual virions of different categories are shown, together with the fields of view from which each is derived.
(EPS)

**S5 Fig. Gallery of selected Tomograms for K25A HIV virions.** Slices through individual virions of different categories are shown, together with the fields of view from which each is derived.
(EPS)

**S6 Fig. IP6-mediated assembly of HIV CA.** (a) Assembly reactions of 75 μM WT CA in buffer containing 50 mM MES pH6, 100 mM NaCl and a range of IP6 concentrations. (b) Negative stain EM images of assembly reactions with 500 μM WT and K25N capsid and 10 μM IP6 in 50 mM MES pH 6.0 40 mM NaCl, Scalebar: 200nm (c) NanoDSF-derived melting

temperatures (Tm's) of WT disulfide-stabilized hexamers or pentamers in the presence of IP6. (EPS)

## Acknowledgments

We acknowledge the MRC—Laboratory of Molecular Biology Electron Microscopy Facility for access and support of electron microscopy sample preparation and data collection and Aaron Tan and Zunlong Ke for help with cryo-ET. We would also like to thank the Diamond light source (proposals mx15916-88 and mx21426-4) and the beamline staff for assistance (Dr Pierre Aller and Mark Williams). We acknowledge use of facilities in the Structural Biology Facility within the Mark Wainwright Analytical Centre–UNSW.

## Author Contributions

**Conceptualization:** Nadine Renner, Donna L. Mallery, Till Böcking, Leo C. James.

**Data curation:** Leo C. James.

**Formal analysis:** Nadine Renner, Donna L. Mallery, K. M. Rifat Faysal, Wang Peng, David A. Jacques, Leo C. James.

**Funding acquisition:** David A. Jacques, Till Böcking, Leo C. James.

**Investigation:** Nadine Renner, Donna L. Mallery, K. M. Rifat Faysal, Wang Peng, David A. Jacques.

**Project administration:** Leo C. James.

**Supervision:** Leo C. James.

**Writing – original draft:** Leo C. James.

**Writing – review & editing:** Nadine Renner, David A. Jacques, Till Böcking, Leo C. James.

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
