## [Decision Letter · Decision Letter 0]

3 Sep 2020

Dear Dr. James,

Thank you very much for submitting your manuscript "A lysine ring in HIV capsid pores coordinates IP6 to drive mature assembly" for consideration at PLOS Pathogens. As with all papers reviewed by the journal, your manuscript was reviewed by members of the editorial board and by several independent reviewers. In light of the reviews (below this email), we would like to invite the resubmission of a significantly-revised version that takes into account the reviewers' comments. We note that while the three reviewers were quite variable in how positively they evaluated this work, all appreciated the interest of the topic. However, reviewer 1, and to some degree reviewer 3, raised concerns that will need to be adequately addressed before we can consider this work for publication. Please note that we will likely ask these two reviewers to re-review the manuscript before a final decision is made..

We cannot make any decision about publication until we have seen the revised manuscript and your response to the reviewers' comments. Your revised manuscript is also likely to be sent to reviewers for further evaluation.

Sincerely,

Bryan R. Cullen

Associate Editor

PLOS Pathogens

Thomas Hope

Section Editor

PLOS Pathogens

Kasturi Haldar

Editor-in-Chief

PLOS Pathogens

orcid.org/0000-0001-5065-158X

Michael Malim

Editor-in-Chief

PLOS Pathogens

orcid.org/0000-0002-7699-2064

Reviewer's Responses to Questions

**Part I - Summary**

Reviewer #1: During particle maturation, pentamers and hexamers of the cleaved capsid (CA) subunit of HIV-1 Gag form a conical lattice around the viral ribonucleoprotein complexes, together forming the viral core. Recent evidence, including work from the PI’s laboratory, indicates that metabolites such as inositol hexakisphosphate (IP6) can stabilize the assembly of the immature and mature HIV-1 CA lattice through interactions between lysine residues present in the pores of CA hexamers (K158 and K227 for immature, and R18 for mature capsid). In this article, Renner et al. perform a number of experiments to investigate the role of IP6 binding to a second ring formed by the K25 residue at the base of the capsid pore in the mature CA hexamer. Based on the crystal structure of the CA hexamer in complex with IP5, the authors identify K25 as a residue potentially involved in coordinating an additional IP6 molecule. As expected from the previously described role of IP6 in vitro in stabilization of the CA lattice, K25A mutation results in severe reverse transcription and replication defects. While the authors propose that the effects of K25A substitution on virus replication is due to the inability to coordinate IP6, the data presented do not fully support this conclusion. For example, in Figure 3, K25A does not prevent binding of IP6 to preassembled hexamers. As it stands, the most prominent effect of K25A substitution appears to be at the level of core assembly and stability. Without a clear demonstration that K25 residue is indeed acting through coordination of IP6 to mediate core assembly and stability, the study fails to add much to our understanding of HIV-1 capsid biology. While the in vitro data in Figure 6 is potentially interesting and supports the conclusion that K25 may stabilize the cores through IP6, it is unclear whether the conditions used in these experiments are physiologically relevant. In addition, proper positive and negative controls seem to be missing in several experiments, which dampened my enthusiasm for this study.

Reviewer #2: In the last years, there has been a revolution on our knowledge about the properties and function the HIV capsid. In 2016, it was discovered that the HIV capsid core contains pores at the capsid-junctions that enable nucleotides to pass through and feed reverse transcription. The presence of a basic ring (R18) in the pore is critical for nucleotide import to occur. These results agreed well with numerous works that in the last decade have shown that the capsid core does not disassemble upon virus entry, but remains fully or partially intact until it reaches the nuclear pore or the nucleus (these two possibilities currently under debate). However, other studies showed that the HIV capsid core is highly unstable, which is a strong counterargument to the intact capsid model. More recently, authors showed that IP molecules (in particular IP5-6) are critical to stabilise the capsid, counteracting the argument of the low HIV capsid core stability. The importance of IPs was restricted to R18 and this study extends it also to K25, suggesting the possibility of a second IP6/5 molecule binding to the pore and playing an important role enabling capsid core assembly. The experiments are well executed and the paper is well written and is easy to follow.

Reviewer #3: Polyanions, in particular IP6, are known to stabilize mature HIV-1 capsids by binding to a positively charged pore at the center of the CA hexamer. James and colleagues have previously shown that a ring made of Arg18 residues, located within the pore, binds IP6 and nucleotides. In the current study, they co-crystallized HIV-1 CA hexamer with two molecules of IP5 (which they use a crystallography-friendly proxy for IP6); and the structure revealed that the CA pore can bind a second polyanion using a ring of Lys25 side chains. Through a series of biochemical and virology experiments they demonstrated the crucial importance of Lys25 to assembly/stability of HIV-1 capsids. Overall, the crystal structure are the results are very interesting and important, although the connection between Lys25 and inositol phosphates remains somewhat tenuous.

**Part II – Major Issues: Key Experiments Required for Acceptance**

Reviewer #1: 1- Figure 1: While IP5 is more practical to be able to obtain a crystal structure, it is unclear whether it really can substitute for IP6. Can the authors minimally show the model of IP6 in this pore and demonstrate that there is no steric hindrance to accommodating an IP6 molecule in that pocket given the axial vs. equatorial orientation of the phosphates? This point should also be clearly discussed in the text.

2- I think the authors should at least aim to separate whether the K25A mutation destabilizes the CA lattice because of the inability to bind IP6 vs. an intrinsic defect in assembly/stability. While I see that this is a difficult chicken-egg type of problem, inclusion of other mutants/controls will strengthen the manuscript and help address this problem. Otherwise, K25 is just another amino acid (in addition many that have already been described in the field) that when mutated destabilizes the core. Thus, I propose:

a. Figure 3a should be expanded to include R18G mutation as a positive control as well as R18G/K25A double mutant (as perhaps the effect will be more apparent once the R18 is mutated?). Other CA destabilizing mutants (i.e. P38A and K203A) should be included as negative controls. In addition, K25A is a pretty drastic mutation and other substitutions such as K25N and K25Q may help dissect the effects on IP6 coordination from the effects on core destabilization.

b. Figure 4 should be expanded to include less drastic K25 substitutions.

c. Figure 6 should be expanded to include less drastic K25 substitutions as well as K203A or P38A substitutions.

3- How do the authors reconcile the seemingly opposite conclusions from Figure 3 and Figure 6? What is the relevance of conditions tested in Figure 6 for HIV-1 biology? How would the result look if a different IP6 concentration was chosen?

4- Results from Figure 3e have inconsistencies. How come R18G can increase the GFP signal at similar levels as K25R?

Reviewer #2: My major concern is about unifying the initial and final experiments regarding the role of IP6 in viral particle formation. If the authors hypothesis is correct and IP5-6 should be bound to K25 to enable maturation, the experiments with purified capsid cores would never work as IP5-6 is added a posteriori to a sample that lacks capsid assemblies (as authors nicely showed). Thus, IP5-6 might be present in the viral particle while this is being formed at the membrane of the cell to enable posterior capsid core assembly during maturation. I wonder if there is any way to test this hypothesis by temporary increasing the levels of IP6/5 in cells (e.g. by overexpression of the IP kinases or by transfection/electroporation of IP6/5?). In this scenario, maybe the proportion of defective particles found in both WT, K25R and/or K25A mutant would be altered (Fig. 3c). Similarly, depletion of IP6/5 in cells should cause an increase of defective particles with Wt virus.

Either way, I think these results should be better explained in the text, so the potential interpretation(S) and derived working models and hypotheses are clearer for a non-expert readers.

Reviewer #3: Thermostability experiments (Fig 3A) seem to suggest that Lys25 may not be critical for IP6 binding to CA hexamers. Based on this, and additional results (specifically the ability of K25A to assemble into tubes but not cones), the authors speculate that the Lys25-IP6 interaction might be more important in the context of the pentamer. A measurement with disulfide-crosslinked pentamers could greatly help support this hypothesis. Further to this, does IP6 fit into the pentamer pore without steric problems?

**Part III – Minor Issues: Editorial and Data Presentation Modifications**

Reviewer #1: 1- Numerous relevant references are missing. For example, there is no reference to the recent work suggesting that the cores may remain intact till they reach to the nucleus. There is no discussion of what is known of core stability and CA mutations involved in maintaining stability.

2- Figure 2c: What is the evidence that chimeric viruses actually form as opposed to having a population of fully WT vs. fully mutant viruses? In addition, there is no reference to Fig. 2c in the text.

3- Figure 2d: Include P38A or K203A substitution in this figure to control for IP6-independent effects in core stability.

4- Figure 2: A figure showing Gag levels/processing in cells and effects of mutations on particle release should be incorporated to this figure.

Reviewer #2: 1. Title: I am not sure that 'mature assembly' is clear for non-expert. You mean 'mature capsid core'?)

2. Abstract. The use of IP5 and IP6 sounds a bit random (although later understandable in the text). Maybe an early clarification that both stabilise the capsid core would be very helpful.

3. Final sentence of intro. It is unclear which of the two models the results provided here support (if they support any). I would be more explicit here.

4. Figure 2c should be referenced in the section describing it.

5. We observed similar dose-dependent decrease [...]. I would clarify that infectivity refers to the purified viral particles.

6. Description of the fluorescent viruses in the main text will be very helpful to understand why and how the fluorescence gets into the particle.

Reviewer #3: (No Response)

PLOS authors have the option to publish the peer review history of their article (what does this mean?). If published, this will include your full peer review and any attached files.

Reviewer #1: No

Reviewer #2: **Yes: **Alfredo Castello

Reviewer #3: No
---

## [Editor Report · Decision Letter 1]

13 Nov 2020

Dear Dr. James,

We are pleased to inform you that your manuscript 'A lysine ring in HIV capsid pores coordinates IP6 to drive mature capsid assembly' has been provisionally accepted for publication in PLOS Pathogens.

Best regards,

Bryan R. Cullen

Associate Editor

PLOS Pathogens

Thomas Hope

Section Editor

PLOS Pathogens

Kasturi Haldar

Editor-in-Chief

PLOS Pathogens

orcid.org/0000-0001-5065-158X

Michael Malim

Editor-in-Chief

PLOS Pathogens

orcid.org/0000-0002-7699-2064
---

## [Editor Report · Acceptance letter]

16 Dec 2020

Dear Dr. James,

We are delighted to inform you that your manuscript, "A lysine ring in HIV capsid pores coordinates IP6 to drive mature capsid assembly," has been formally accepted for publication in PLOS Pathogens.

Best regards,

Kasturi Haldar

Editor-in-Chief

PLOS Pathogens

orcid.org/0000-0001-5065-158X

Michael Malim

Editor-in-Chief

PLOS Pathogens

orcid.org/0000-0002-7699-2064